# HyperEmbed: Tradeoffs Between Resources and Performance in NLP Tasks with Hyperdimensional Computing enabled embedding of $n$-gram statistics

## Abstract

Recent advances in Deep Learning have led to a significant performance increase on several NLP tasks, however, the models become more and more computationally demanding. Therefore, this paper tackles the domain of computationally efficient algorithms for NLP tasks. In particular, it investigates distributed representations of $n$-gram statistics of texts. The representations are formed using hyperdimensional computing enabled embedding. These representations then serve as features, which are used as input to standard classifiers. We investigate the applicability of the embedding on one large and three small standard datasets for classification tasks using nine classifiers. The embedding achieved on par $F_1$ scores while decreasing the time and memory requirements by several times compared to the conventional $n$-gram statistics, e.g., for one of the classifiers on a small dataset, the memory reduction was $6.18$ times; while train and test speed-ups were $4.62$ and $3.84$ times, respectively. For many classifiers on the large dataset, the memory reduction was about $100$ times and train and test speed-ups were over $100$ times. More importantly, the usage of distributed representations formed via hyperdimensional computing allows dissecting the strict dependency between the dimensionality of the representation and the parameters of $n$-gram statistics, thus, opening a room for tradeoffs.

## 1 Introduction

Recent work (Strubell et al., 2019) has brought significant attention by demonstrating potential cost and environmental impact of developing and training state-of-the-art models for Natural Language Processing (NLP) tasks. The work suggested several countermeasures for changing the situation. One of them recommends a concerted effort by industry and academia to promote research of more computationally efficient algorithms. The main focus of this paper falls precisely in this domain.

In particular, we consider NLP systems using a well-known technique called n-gram statistics. The key idea is that hyperdimensional computing (Kanerva, 2009) allows forming distributed representations of the conventional n-gram statistics (Joshi et al., 2016). The use of these distributed representations, in turn, allows trading-off the performance of an NLP system (e.g., $F_1$ score) and its computational resources (i.e., time and memory). The main contribution of this paper is the systematic study of these tradeoffs on nine machine learning algorithms using several benchmark classification datasets. We demonstrate the usefulness of hyperdimensional computing-based embedding, which is highly time and memory efficient. Our experiments on a well-known dataset (Braun et al., 2017) for intent classification show that it is possible to reduce memory usage by $\sim 10x$ and speed-up training by $\sim 5x$ without compromising the $F_1$ score. Several important use-cases are motivating the efforts towards trading-off the performance of a system against computational resources required to achieve that performance: high-throughput systems with an extremely large number of requests/transactions (the power of one per cent); resource-constrained systems where computational resources and energy are scarce (edge computing); green computing systems taking into account the aspects of environmental sustainability when considering the efficiency of algorithms (AI HLEG, 2019).

The paper is structured as follows. Section 2 covers the related work. Section 3 outlines the evaluation and describes the datasets. The methods being used are presented in Section 4. Section 5 evaluates of the experimental results. Discussion and concluding remarks are presented in Section 6.

## 2 RELATED WORK

Commonly, data for NLP tasks are represented in the form of vectors, which are then used as an input to machine learning algorithms. These representations range from dense learnable vectors to extremely sparse non-learnable vectors. Well-known examples of such representations include one-hot encodings, count-based vectors, and Term Frequency Inverse Document Frequency (TF-IDF) among others. Despite being very useful, non-learnable representations have their disadvantages such as resource inefficiency due to their sparsity and absence of contextual information (except for TF-IDF). Learnable vector representations such as word embeddings (e.g., Word2Vec (Mikolov et al., 2013) or GloVe (Pennington et al., 2014)) partially address these issues by obtaining dense vectors in an unsupervised learning fashion. These representations are based on the distributional hypothesis: words located nearby in a vector space should have similar contextual meaning. The idea has been further improved in Joulin et al. (2016) by representing words with character $n$-grams. Another efficient way of representing a word is the concept of Byte Pair Encoding, which has been introduced in Gage (1994). The disadvantage of the learnable representations, however, is that they require pretraining involving large train corpus as well as have a large memory footprint (in order of GB). As an alternative to word/character embedding, Shridhar et al. (2019) introduced the idea of Subword Semantic Hashing that uses a hashing method to represent subword tokens, thus, reducing the memory footprint (in order of MB) and removing the necessity of pretraining over a large corpus. The approach has demonstrated the state-of-the-art results on three datasets for intent classification.

The Subword Semantic Hashing, however, relies on $n$-gram statistics for extracting the representation vector used as an input to classification algorithms. It is worth noting that the conventional $n$-gram statistics uses a positional representation where each position in the vector can be attributed to a particular $n$-gram. The disadvantage of the conventional $n$-gram statistics is that the size of the vector grows exponentially with $n$. Nevertheless, it is possible to untie the size of representation from $n$ by using distributed representations (Hinton et al., 1986), where the information is distributed across the vectors positions. In particular, Joshi et al. (2016) suggest how to embed conventional $n$-gram statistics into a high-dimensional vector (HD vector) using the principles of hyperdimensional computing. Hyperdimensional computing also known as Vector Symbolic Architectures (Plate, 2003; Kanerva, 2009; Eliasmith, 2013) is a family of bio-inspired methods of manipulating and representing information. The method of embedding $n$-gram statistics into the distributed representation in the form of an HD vector has demonstrated promising results on the task of language identification while being hardware-friendly (Rahimi et al., 2016). In Najafabadi et al. (2016) it was further applied to the classification of news articles into one of eight predefined categories. The method has also shown promising results (Kleyko et al., 2019) when using HD vectors for training Self-Organizing Maps (Kohonen, 2001). However there are no previous studies comprehensively exploring tradeoffs achievable with the method on benchmark NLP datasets when using the supervised classifiers.

## 3 EVALUATION OUTLINE

### 3.1 CLASSIFIERS AND PERFORMANCE METRICS

To obtain the results applicable to a broad range of existing machine learning algorithms, we have performed experiments with several conventional classifiers. In particular, the following classifiers were studied: Ridge Classifier, k-Nearest Neighbors (kNN), Multilayer Perceptron (MLP), Passive Aggressive, Random Forest, Linear Support Vector Classifier (SVC), Stochastic Gradient Descent (SGD), Nearest Centroid, and Bernoulli Naive Bayes (NB). All the classifiers are available in the scikit-learn library (Pedregosa et al., 2011), which was used in the experiments.

Since the main focus of this paper is the tradeoff between classification performance and computational resources, we have to define metrics for both aspects. The quality of the classification performance of a model will be measured by a simple and well-known metric – $F_1$ score (please

see (Fawcett, 2006)). The computational resources will be characterized by three metrics: the time it takes to train a model, the time it takes to test the trained model, and the memory, where the memory is defined as the sum of the size of input feature vectors for train and test splits as well as the size of the trained model. To avoid the dependencies such as particular specifications of a computer and dataset size, the train/test times and memory are reported as relative values (i.e., train/test speed-up and memory reduction), where the reference is the value obtained for the case of the conventional $n$-gram statistics.[1]

## 3.2 DATASETS

Four different datasets were used to obtain the empirical results reported in this paper: the *Chatbot Corpus* (Chatbot), the *Ask Ubuntu Corpus* (AskUbuntu), the *Web Applications Corpus* (WebApplication), and the *20 News Groups Corpus* (20NewsGroups). The first three are referred to as small datasets. The Chatbot dataset comprises questions posed to a Telegram chatbot. The chatbot, in turn, replied the questions of the public transport of Munich. The AskUbuntu and WebApplication datasets are questions and answers from the StackExchange. The 20NewsGroups dataset comprises news posts labelled into several categories. All datasets have predetermined train and test splits. The first three datasets (Braun et al., 2017) are available on GitHub.[2]

The Chatbot dataset consists of two intents: the *(Departure Time and Find Connection)* with 206 questions. The corpus has a total of five different entity types *(StationStart, StationDest, Criterion, Vehicle, Line)*, which were not used in our benchmarks, as the results were only for intent classification. The samples come in English. Despite this, the train station names are in German, which is evident from the text where the German letters appear (ä,ö,ü,ß). The dataset has the following data sample distribution (train/test): Departure Time ($43/35$); Find Connection ($57/71$).

The AskUbuntu dataset comprises five intents with the following data sample distribution (train/test): Make Update ($10/37$); Setup Printer ($10/13$); Shutdown Computer ($13/14$); Software Recommendation ($17/40$); None ($3/5$). Thus, it includes 162 samples in total. The samples were gathered directly from the AskUbuntu platform. Only questions with the highest scores and upvotes were considered. For the task of mapping the correct intent to the question, the Amazon Mechanical Turk was employed. Beyond the questions labelled with their intent, this dataset contains also some extra information such as author, page URL with the question, entities, answer, and the answer's author. It is worth noting that none of these data was used in the experiments.

The WebApplication dataset comprises 89 text samples of eight intents with the following distribution (train/test): Change Password ($2/6$); Delete Account ($7/10$); Download Video ($1/0$); Export Data ($2/3$); Filter Spam ($6/14$); Find Alternative ($7/16$); Sync Accounts ($3/6$); None ($2/4$).

The 20NewsGroups dataset has been originally collected by Ken Lang. It comprises of 20 categories (for details please see Table 7 in the Appendix). Each category has exactly $18,846$ text samples. Moreover, the samples of each category are split neatly into the train ($11,314$ samples) and test ($7,532$ samples) sets. The dataset comes already prepackaged with the *scikitlearn* library for Python.

## 4 METHODS

### 4.1 CONVENTIONAL $n$-GRAM STATISTICS

An empty vector $s$ stores $n$-gram statistics for an input text $\mathcal{D}$. $\mathcal{D}$ consists of symbols from the alphabet of size $a$; $i$th position in $s$ keeps the counter of the corresponding $n$-gram $\mathbb{A}_i = \langle \mathcal{S}_1, \mathcal{S}_2, \ldots, \mathcal{S}_n, \rangle$ from the set $\mathbb{A}$ of all unique $n$-grams; $\mathcal{S}_j$ corresponds to a symbol in $j$th position of $\mathbb{A}_i$. The dimensionality of $s$ equals the total number of $n$-grams in $\mathbb{A}$ and calculated as $a^n$. Usually, $s$ is obtained via a single pass-through $\mathcal{D}$ using the overlapping sliding window of size $n$. The value of a position in $s$ (i.e., counter) corresponding to a $n$-gram observed in the current window is incremented by one. In other words, $s$ summarizes how many times each $n$-gram in $\mathbb{A}$ was observed in $\mathcal{D}$.

---

[1] It is worth noting that the speed-ups reported in Section 5 do not include the time it takes to obtain the corresponding HD vectors. Please see the discussion of this issue in Section 6.

[2] Under the Creative Commons CC BY-SA 3.0 license: `https://github.com/sebischair/NLU-Evaluation-Corpora`

### 4.2 WORD EMBEDDINGS WITH SUBWORD INFORMATION

Work by Bojanowski et al. (2017) demonstrated that words' representations can be formed via learning character $n$-grams, which are then summed up to represent words. This method (FastText) has an advantage over the conventional word embeddings since unseen words could be better approximated as it is highly likely that some of their $n$-gram subwords have already appeared in other words. Therefore, each word $w$ is represented as a bag of its character $n$-gram. Special boundary symbols "<" and ">" are added at the beginning and the end of each word. The word $w$ itself is added to the set of its $n$-grams, to learn a representation for each word along with character $n$-grams. Taking the word *have* and $n = 3$ as an example, $(have) = [< ha, hav, ave, ve >, have]$. Formally, for a given word $w$, $\mathcal{N}_w \subset \{1, \dots, G\}$ denotes the set of $G$ $n$-grams appearing in $w$. Each $n$-gram $g$ has an associated vector representation $\boldsymbol{z}_g$. Word $w$ is represented as the sum of the vector representations of its $n$-grams. A scoring function $g$ is defined for each word that is represented as a set of respective $n$-grams and the context word (denoted as $c$), as:

$$g(w, c) = \sum_{g \in \mathcal{N}_w} \boldsymbol{z}_g^\top \boldsymbol{v}_c,$$

where $\boldsymbol{v}_c$ is the vector representation of the context word $c$. Practically, a word is represented by its index in the word dictionary and a set of $n$-grams it contains.

### 4.3 BYTE PAIR ENCODING

The idea of Byte Pair Encoding (BPE) was introduced in Gage (1994). BPE iteratively replaces the most frequent pair of bytes in a sequence with a single, unused byte. It can be similarly used to merge characters or character sequences for words representations. A symbol vocabulary is initialized with a character vocabulary with every word represented in the form of characters, where "." is used as the end of word symbol. All symbol pairs are counted iteratively and then replaced with a new symbol. Each operation results in a new symbol, which represents an $n$-gram. Similarly, frequently occurring $n$-grams are eventually merged into a single symbol. This makes the final vocabulary size equal to the sum of initial vocabulary and number of merge operations.

### 4.4 SUBWORD SEMANTIC HASHING

Subword Semantic Hashing (SemHash) is described in details in Shridhar et al. (2019); Huang et al. (2013). SemHash represents the input sentence in the form of subword tokens using a hashing method reducing the collision rate. These subword tokens act as features to the model and can be used as an alternative to word/$n$-gram embeddings. For a given input sample text $T$, e.g., *"I have a flying disk"*, we split it into a list of words $t_i$. The output of the split would look as follows: *["I", "have", "a", "flying", "disk"]*. Each word is then passed into a prehashing function $\mathcal{H}(t_i)$. $\mathcal{H}(t_i)$ first adds a # at the beginning and at the end of $t_i$. Then it generates subwords via extracting $n$-grams ($n$=3) from #$t_i$#, e.g., $\mathcal{H}(have) = [\#ha, hav, ave, ve\#]$. These tri-grams are the subwords denoted as $t_i^j$, where $j$ is the index of a subword. $\mathcal{H}(t_i)$ is then applied to the entire text corpus to generate subwords via $n$-gram statistics. These subwords are used to extract features for a given text.

### 4.5 EMBEDDING $n$-GRAM STATISTICS INTO AN HD VECTOR

Alphabet's symbols are the most basic elements of a system. We assign each symbol with a random $d$-dimensional bipolar HD vector. These vectors are stored in a matrix (denoted as $\boldsymbol{H}$, where $\boldsymbol{H} \in [d \times a]$), which is referred to as the item memory, For a given symbol $\mathcal{S}$ its HD vector is denoted as $\boldsymbol{H}_\mathcal{S} \in \{-1, +1\}^{[d \times 1]}$. To manipulate HD vectors, hyperdimensional computing defines three key operations[3] on them: bundling[4] (denoted with + and implemented via position-wise addi-

---

[3] Please see Kanerva (2009) for proper definitions and properties of hyperdimensional computing operations.

[4] The bundling operation allows storing information in HD vectors Frady et al. (2018); if several copies of any HD vector are included (e.g., $2\boldsymbol{H}_{\mathcal{S}_1} + \boldsymbol{H}_{\mathcal{S}_2}$), the resultant HD vector is more similar to the dominating HD vector than to other components. Since this paper does not go into deep analytical details of why HD vectors allow embedding the conventional n-gram statistics, the diligent readers are referred to Frady et al. (2018) for the relevant analysis.

tion), binding (denoted with $\odot$ and implemented via position-wise multiplication), and permutation[5] (denoted with $\rho$).

Three operations above allow embedding $n$-gram statistics into distributed representation (HD vector) Joshi et al. (2016). First, $\boldsymbol{H}$ is generated for the alphabet. A position of symbol $\mathcal{S}_j$ in $\mathbb{A}_i$ is represented by applying $\rho$ to the corresponding HD vector $\boldsymbol{H}_{\mathcal{S}_j}$ $j$ times, which is denoted as $\rho^j(\boldsymbol{H}_{\mathcal{S}_j})$. Next, a single HD vector for $\mathbb{A}_i$ (denoted as $\boldsymbol{m}_{\mathbb{A}_i}$) is formed via the consecutive binding of permuted HD vectors $\rho^j(\boldsymbol{H}_{\mathcal{S}_j})$ representing symbols in each position $j$ of $\mathbb{A}_i$. For example, the trigram 'cba' will be mapped to its HD vector as follows: $\rho^1(\boldsymbol{H}_\mathrm{c}) \odot \rho^2(\boldsymbol{H}_\mathrm{b}) \odot \rho^3(\boldsymbol{H}_\mathrm{a})$. In general, the process of forming HD vector of an $n$ can be formalized as follows:

$$\boldsymbol{m}_{\mathbb{A}_i} = \prod_{j=1}^{n} \rho^j(\boldsymbol{H}_{\mathcal{S}_j}),$$

where $\prod$ denotes the binding operation when applied to $n$ HD vectors. Once it is known how to get $\boldsymbol{m}_{\mathbb{A}_i}$, embedding the conventional $n$-gram statistics stored in $\boldsymbol{s}$ (see section 4.1) is straightforward. HD vector $\boldsymbol{h}$ corresponding to $\boldsymbol{s}$ is created by bundling together all $n$-grams observed in the data:

$$\boldsymbol{h} = \sum_{i=1}^{a^n} \boldsymbol{s}_i \boldsymbol{m}_{\mathbb{A}_i} = \sum_{i=1}^{a^n} \boldsymbol{s}_i \prod_{j=1}^{n} \rho^j(\boldsymbol{H}_{\mathcal{S}_j}),$$

where $\sum$ denotes the bundling operation when applied to several HD vectors. Note that $\boldsymbol{h}$ is not bipolar, therefore, in the experiments below we normalized it by its $\ell_2$ norm.

## 5 EMPIRICAL EVALUATION

### 5.1 SETUP

All datasets were preprocessed by removing control characters and stop words (see the Appendix for details). It is also worth noting that the realization of the conventional $n$-gram statistics used in the experiments was forming a model, which was storing only $n$-grams present in the train split.

The range of $n$ in the experiments with small datasets was $[2-4]$ while for the 20NewsGroups dataset it was $[2-3]$ since the number of possible 4-grams was overwhelming. All results reported for small datasets were obtained by averaging across 50 independent simulations. In the case of the 20NewsGroups dataset, the number of simulations was decreased to 10 due to high computational costs. To have a fair comparison of computational resources, all results for small datasets were obtained on a dedicated laptop without involving GPUs while the results for the 20NewsGroups dataset were obtained with a computing cluster (CPU only) without the intervention of other users. More details about experimental settings could be found in the Appendix section.

### 5.2 RESULTS

First, we report the results of the MLP classifier on all datasets as it represents a widely used class of algorithms – neural networks. The goal of the experiments was to observe how the dimensionality of HD vectors embedding $n$-gram statistics affects the $F_1$ scores and the computational resources. Figures 1a-1d present the results for the AskUbuntu, Chatbot, WebApplication, and 20NewsGroups datasets, respectively. The dimensionality of HD vectors varied as $2^k$, $k \in [5, 14]$. All figures have an identical structure. Shaded areas depict 95% confidence intervals. Left panels depict the $F_1$ score while right panels depict the train and test speed-ups as well as memory reduction. Note that there are different scales ($y$-axes) in the right panels. A solid horizontal line indicates 1 for the corresponding $y$-axis, i.e., the moment when both models consume the same resources.

The results in all figures are consistent in a way that up to a certain point $F_1$ score was increasing with the increasing dimensionality. For the small datasets even small dimensionalities of HD vectors (e.g., $32 = 2^5$) led to the $F_1$ scores, which are far beyond random. For example, for the AskUbuntu dataset, it was $84\%$ of the conventional $n$-gram statistics $F_1$ score. For the values above 512 the

---

[5]It is convenient to use $\rho$ to bind symbol's HD vector with its position in a sequence.

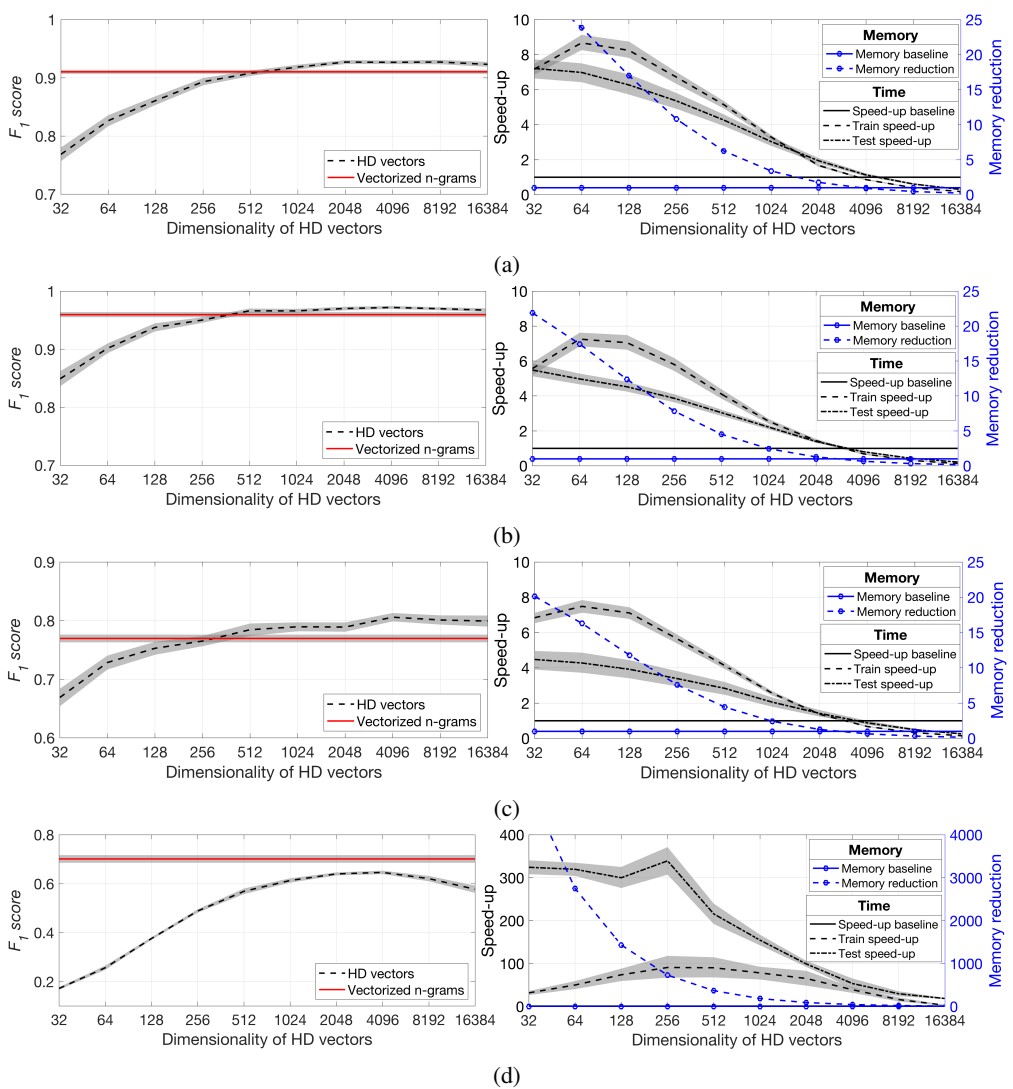

Figure 1: MLP results vs. the dimensionality of HD vectors on: (a) the AskUbuntu dataset. (b) the Chatbot dataset. (c) the WebApplication dataset. (d) the 20NewsGroups dataset.

performance saturation begins. Moreover, the improvements beyond 2048 are marginal. The situation is more complicated for the 20NewsGroups dataset where for 32-dimensional HD vectors $F_1$ score is fairly low though still better than a random guess (0.05). However, it increases steeply until 1024 and achieves its maximum at 4096 being 92 % of the conventional $n$-gram statistics $F_1$ score. The dimensionalities above 4096 showed worse results.

When it comes to computational resources, there is a similar pattern for all the datasets. The train/test speed-ups and memory reduction are diminishing with the increased dimensionality of HD vectors. At the point when the dimensionality of HD vectors equals the size of the conventional $n$-gram statistics, both approaches consume approximately the same resources. These points in the figures are different because the datasets have different size of $n$-gram statistics: 3729, 2753, 2734, and 192652, for the AskUbuntu, Chatbot, WebApplication, and 20NewsGroups datasets, respectively. Also, for all datasets, the memory reduction is higher than the speed-ups. The most impressive speed-ups and reductions were observed for the 20NewsGroups dataset (e.g., 186 times less memory for 1024-dimensional HD vectors). This is due to its large size it contains a huge number of $n$-grams resulting in large size of the $n$-gram statistics. Nevertheless, even for small datasets, the gains were noticeable. For instance, for the WebApplication dataset at 256 $F_1$ score was 99 % of the

Table 1: Performance of all classifiers for the AskUbuntu dataset.

| Classifier | $F_1$ score | | | Resources: SH vs. HD | | | Resources: SH vs. BPE | | |
|---|---|---|---|---|---|---|---|---|---|
| | SH | BPE | HD | Tr. | Ts. | Mem. | Tr. | Ts. | Mem. |
| MLP | 0.92 | 0.91 | 0.91 | 4.62 | 3.84 | 6.18 | 1.67 | 1.61 | 1.72 |
| Passive Aggr. | 0.92 | 0.93 | 0.90 | 4.86 | 3.07 | 6.31 | 2.19 | 2.14 | 1.76 |
| SGD Classifier | 0.89 | 0.89 | 0.88 | 4.66 | 3.50 | 6.31 | 1.94 | 2.16 | 1.76 |
| Ridge Classifier | 0.90 | 0.91 | 0.90 | 3.91 | 4.74 | 6.31 | 1.63 | 1.62 | 1.76 |
| KNN Classifier | 0.79 | 0.72 | 0.82 | 2.11 | 4.53 | 8.48 | 1.56 | 1.79 | 1.76 |
| Nearest Centroid | 0.90 | 0.89 | 0.90 | 1.66 | 3.41 | 6.32 | 1.35 | 1.87 | 1.76 |
| Linear SVC | 0.90 | 0.92 | 0.90 | 1.18 | 2.39 | 6.29 | 0.91 | 1.91 | 1.76 |
| Random Forest | 0.88 | 0.90 | 0.86 | 0.91 | 1.09 | 6.11 | 1.15 | 0.96 | 1.75 |
| Bernoulli NB | 0.91 | 0.92 | 0.85 | 2.30 | 3.72 | 6.34 | 1.96 | 2.42 | 1.76 |

Table 2: Performance of all classifiers for the Chatbot dataset.

| Classifier | $F_1$ score | | | Resources: SH vs. HD | | | Resources: SH vs. BPE | | |
|---|---|---|---|---|---|---|---|---|---|
| | SH | BPE | HD | Tr. | Ts. | Mem. | Tr. | Ts. | Mem. |
| MLP | 0.96 | 0.94 | 0.96 | 3.42 | 2.62 | 4.58 | 1.86 | 1.52 | 1.86 |
| Passive Aggr. | 0.95 | 0.91 | 0.94 | 4.40 | 2.38 | 4.72 | 2.29 | 2.22 | 1.92 |
| SGD Classifier | 0.93 | 0.93 | 0.92 | 3.16 | 2.06 | 4.72 | 1.88 | 1.84 | 1.92 |
| Ridge Classifier | 0.94 | 0.94 | 0.92 | 2.88 | 2.22 | 4.72 | 1.67 | 1.38 | 1.92 |
| KNN Classifier | 0.75 | 0.71 | 0.83 | 1.66 | 3.59 | 6.51 | 1.43 | 1.79 | 1.92 |
| Nearest Centroid | 0.89 | 0.94 | 0.84 | 1.41 | 2.13 | 4.73 | 1.17 | 1.61 | 1.92 |
| Linear SVC | 0.94 | 0.93 | 0.94 | 0.52 | 1.57 | 4.72 | 1.28 | 1.66 | 1.92 |
| Random Forest | 0.95 | 0.95 | 0.91 | 0.95 | 1.10 | 4.61 | 1.16 | 0.98 | 1.91 |
| Bernoulli NB | 0.93 | 0.93 | 0.82 | 1.92 | 2.60 | 4.73 | 1.53 | 1.72 | 1.92 |

conventional $n$-gram statistics while the train/test speed-ups and the memory reduction were 5.6, 3.4, and 7.6, respectively.

Thus, these empirical results suggest that the quality of embedding w.r.t. the achievable $F_1$ score improves with increased dimensionality, however, after a certain saturation or peak point increasing dimensionality further either does not affect or worsen the classification performance and arguably becomes impractical when considering the computational resources.

Tables 1-3[6] report the results for three small datasets[7] when applying all the considered classifiers. Due to the space limitation, a fixed dimensionality of HD vectors is reported only: 512 for small datasets in Tables 1-3 and 2048 for the 20NewsGroups dataset in Table 9. These dimensionalities were chosen based on the results in Figures 1a-1d as the ones allowing to achieve a good approximation of $F_1$ score while providing substantial speed-up/reduction. We also performed experiments when using the BPE instead of the SemHash before extracting $n$-gram statistics. Throughout the tables, the BPE demonstrated $F_1$ scores comparable to that of the SemHash while showing the train/test speed-ups and memory reduction at about 2. This is because the usage of the BPE resulted in smaller sizes of the $n$-gram statistics, which were 2176, 1467, and 1508 for the AskUbuntu, Chatbot, and WebApplication datasets, respectively.

In the case of HD vectors, the picture is less coherent. For example, there is a group of classifiers (e.g., MLP, SGD, KNN) where $F_1$ scores are well approximated (or even improved) while achieving noticeable computational reductions. In the case of Linear SVC, $F_1$ scores are well-preserved and there is $4-6$ memory reduction but test/train speed-ups are marginal (even slower for training the Chatbot). This is because Linear SVC implementation benefits from sparse representations (con-

---

[6]The notations Tr., Ts., Mem. in the tables stand for the train speed-up, test speed-up, and the memory reduction for the given classifier, respectively. SH stands for SemHash.

[7] Due to the page limit, results for the 20NewsGroups dataset are presented in the Appendix (see Table 9).

Table 3: Performance of all classifiers for the WebApplication dataset.

| Classifier | $F_1$ score | | | Resources: SH vs. HD | | | Resources: SH vs. BPE | | |
|---|---|---|---|---|---|---|---|---|---|
| | SH | BPE | HD | Tr. | Ts. | Mem. | Tr. | Ts. | Mem. |
| MLP | 0.77 | 0.77 | 0.79 | 3.10 | 2.00 | 4.43 | 1.74 | 1.44 | 1.73 |
| Passive Aggr. | 0.82 | 0.80 | 0.80 | 3.73 | 1.45 | 4.33 | 1.86 | 1.32 | 1.75 |
| SGD Classifier | 0.75 | 0.74 | 0.73 | 3.01 | 1.87 | 4.33 | 1.62 | 1.32 | 1.75 |
| Ridge Classifier | 0.79 | 0.80 | 0.80 | 1.66 | 2.40 | 4.34 | 0.71 | 1.09 | 1.75 |
| KNN Classifier | 0.72 | 0.75 | 0.76 | 1.16 | 2.76 | 5.96 | 1.14 | 1.51 | 1.76 |
| Nearest Centroid | 0.74 | 0.73 | 0.77 | 1.42 | 1.79 | 4.34 | 1.13 | 1.21 | 1.75 |
| Linear SVC | 0.82 | 0.80 | 0.80 | 1.04 | 1.48 | 4.29 | 0.47 | 1.18 | 1.75 |
| Random Forest | 0.87 | 0.85 | 0.72 | 0.95 | 1.26 | 4.11 | 1.05 | 1.12 | 1.73 |
| Bernoulli NB | 0.74 | 0.75 | 0.64 | 1.51 | 2.08 | 4.38 | 1.19 | 1.49 | 1.75 |

ventional n-gram statistics) while HD vectors in this study are dense. Last, for Bernoulli NB and Random Forest $F_1$ scores were not approximated well (cf. 0.93 vs. 0.82 for Bernoulli NB in the case of the Chatbot). This is likely because both classifiers are relying on local information contained in individual features, which is not the case in HD vectors where information is represented distributively across the whole vector. The slow train time of Random Forest is likely because in the absence of well-separable features it tries to construct large trees.

Last, due to the difference in the implementation (the official implementation of FastText only uses a linear classifier), we were not able to have a proper comparison of computational resources with the FastText.[8] However, we obtained the following $F_1$ scores with auto hyperparameter search: 0.91, 0.97, 0.76 for the AskUbuntu, Chatbot, and WebApplication datasets, respectively. These results indicate that for the considered datasets there is no drastic classification performance improvement (even worse for the WebApplication) when using the learned representations of $n$-grams.

## 6 DISCUSSION AND CONCLUSIONS

The first observation is that the results on the 20NewsGroups dataset are not the state-of-the-art, which is currently 0.92 $F_1$ score achieved with the BERT model as reported in Mahabal et al. (2019). Nevertheless, it is important to keep in mind that the main goal of the experiments with the 20NewsGroups dataset has been to demonstrate that $n$-gram statistics embedded into HD vectors allows getting the tradeoff even for a large text corpus. We even observed that for large datasets the usage of HD vectors is likely to provide the best gains in terms of resource-efficiency. Moreover, the gains on the small datasets were also noteworthy (several times). Thus, based on these observations we conclude that HyperEmbed would be a very useful feature in the standard ML libraries. A more general conclusion is that it is worth revisiting results in the area of random projection (Rachkovskij, 2016) as they are likely to allow achieving performance/resources tradeoff in a range of NLP scenarios.

It was stated in Section 3.1 the speed-ups reported above did not include the time for forming HD vectors. The main reason for that is that our Python-based implementation of the method was quite inefficient, especially the cyclic shifts implemented with numpy.roll. At the same time, as it could be seen from the formulation of the embedding method in Section 4.5 its complexity is linear and depends on $n$ as well as on the length of the sample text, thus, fast implementation is doable. We made the proof-of-concept implementation in Matlab, which is much faster. For example, for the AskUbuntu dataset forming 512-dimensional HD vectors of the train split (the same machine) took about 7.5 % of the MLP training time, which is a positive result.

Despite the demonstrated tradeoffs between the $F_1$ score and the computational resources, it is extremely hard to have an objective function, which would tell us when the compromise is acceptable and when it is not. In our opinion, a general solution would be to define a utility function, which

---

[8]We could have implemented the algorithm ourselves but it can be claimed unfair to compare the required memory and time, if we do not use the best practices, which are unknown to us.

would be able to assign a certain cost to both a unit of performance (e.g., 0.01 increase in $F_1$ score) and a unit of computation (e.g., 10 % decrease in the inference time). The use of the utility function would allow deciding whether an alternative solution, which is, e.g., faster but less accurate, is better or not than the existing one. However, the main challenge here would be to define such a utility function since it would have to be defined for each particular application. Moreover, defining such functions even for the considered classification problems is out of the scope of this study. Nevertheless, we believe that it is the way forward to get an objective comparison criterion.

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

## A    EXPERIMENTAL SETTINGS

**Preprocessing**    All datasets were preprocessed using the spacy library. It was used to remove stop words from data. We used the spacy model called "en_core_web_lg" to parse the datasets. Last, all text samples were preprocessed by removing control characters; in particular, the ones in the set *[Cc]*, which includes Unicode characters from *U+0000* to *U+009F*.

Since the 20NewsGroups dataset is already large, it does not seem to be necessary to apply the SemHash to it, therefore, it was omitted in the experiments (i.e., SH in Table 9 refers to pure $n$-grams). Last, the small datasets were augmented, making all smaller classes having the same number of samples as the largest class in the train split for that dataset. Using WordNet as a dictionary, nouns and verbs were swapped with their synonyms creating new sentences until all the classes for that set have the same number of samples. The final distributions are shown in Tables 4–6.

**Byte Pair Encoding**    Vocabulary Size of 1000 was used for WebApplication and AskUbuntu dataset whereas a vocabulary size of 250 was used for Chatbot dataset due to its smaller size. N-gram range of (2-4) was used with analyzer as *char*. Cross Validation was set to 5.

**FastText**    Autotune Validation was used to find the optimal hyperparameters for all the dataset. No quantization of the model was performed to prevent the compromise on model accuracy.

**Hyperparameters**    In order to find optimal hyperparameters, a grid-based search was applied to three small datasets for the following classifiers: MLP, Random Forest, and KNN. The configuration performing best among all small datasets was chosen to be used in order to report the results reported in the paper. Moreover, the same configuration was used for the 20NewsGroups dataset. In the case of MLP, four different configurations of hidden layers were considered: [(100, 50), (300, 100),(300, 200, 100), and (300, 100, 50)]; (300, 100, 50) configuration has been chosen. The maximal number of MLP iterations was set to 500. In the case of Random Forest, two hyperparameters were optimized number of estimators ([50, 60, 70]) and minimum samples leaf ([1, 11]); we used 50 estimators and 1 leaf. In the case of KNN, the number of neighbors between 3 and 7 was considered; 3 neighbors were used in the experiments. For all the other classifiers the default hyperparameter settings provided by Sklearn library were used.

## B    DISTRIBUTION OF THE DATASETS

Tables 4–7 show the data distribution of the four datasets used in the empirical evaluation.

Table 4: Data sample distribution for the Chatbot dataset

| Intent | Train original | Train Augmented | Test |
|---|---|---|---|
| Departure Time | 43 | 57 | 35 |
| Find Connection | 57 | 57 | 71 |

Table 5: Data sample distribution for the AskUbuntu dataset

| Intent | Train original | Train Augmented | Test |
|---|---|---|---|
| Make Update | 10 | 17 | 37 |
| Setup Printer | 10 | 17 | 13 |
| Shutdown Computer | 13 | 17 | 14 |
| Software Recommendation | 17 | 17 | 40 |
| None | 3 | 17 | 5 |

Table 6: Data sample distribution for the WebApplication dataset

| Intent | Train original | Train Augmented | Test |
|---|---|---|---|
| Change Password | 2 | 7 | 6 |
| Delete Account | 7 | 7 | 10 |
| Download Video | 1 | 7 | 0 |
| Export Data | 2 | 7 | 3 |
| Filter Spam | 6 | 7 | 14 |
| Find Alternative | 7 | 7 | 16 |
| Sync Accounts | 3 | 7 | 6 |
| None | 2 | 7 | 4 |

Table 7: Data sample distribution for the 20NewsGroups dataset

| Categories | Train | Test |
|---|---|---|
| alt.atheism | 11314 | 7532 |
| comp.graphics | 11314 | 7532 |
| comp.os.ms-windows.misc | 11314 | 7532 |
| comp.sys.ibm.pc.hardware | 11314 | 7532 |
| comp.sys.mac.hardware | 11314 | 7532 |
| comp.windows.x | 11314 | 7532 |
| misc.forsale | 11314 | 7532 |
| rec.autos | 11314 | 7532 |
| rec.motorcycles | 11314 | 7532 |
| rec.sport.baseball | 11314 | 7532 |
| rec.sport.hockey | 11314 | 7532 |
| sci.crypt | 11314 | 7532 |
| sci.electronics | 11314 | 7532 |
| sci.electronics | 11314 | 7532 |
| sci.space | 11314 | 7532 |
| soc.religion.christian | 11314 | 7532 |
| talk.politics.guns | 11314 | 7532 |
| talk.politics.mideast | 11314 | 7532 |
| talk.politics.misc | 11314 | 7532 |
| talk.religion.misc | 11314 | 7532 |

## C  ADDITIONAL RESULTS NOT INCLUDED IN THE MAIN TEXT

One thing to note in Table 8 is the differences in the $F_1$ scores of the Semantic Hashing approach from the ones reported in Shridhar et al. (2019) for all three small datasets. There were some data augmentation techniques, which were used in the paper, most prominently a QWERTY-based word augmentation accounting for the spelling mistakes. This technique was not used in this work, which resulted in a slight difference in the obtained $F_1$ scores.

Note also that Table 9 does not report the results for the BPE. This is purely due to high computational costs required to obtain the BPE model and vocabulary.

Table 8: $F_1$ score comparison of various platforms on three smaller datasets with methods mentioned in the paper. Some results are taken from Shridhar et al. (2019)

| Platform | Chatbot | AskUbuntu | WebApp | Average |
|---|---|---|---|---|
| Botfuel | 0.98 | 0.90 | 0.80 | 0.89 |
| Luis | 0.98 | 0.90 | 0.81 | 0.90 |
| Dialogflow | 0.93 | 0.85 | 0.80 | 0.86 |
| Watson | 0.97 | 0.92 | 0.83 | 0.91 |
| Rasa | 0.98 | 0.86 | 0.74 | 0.86 |
| Snips | 0.96 | 0.83 | 0.78 | 0.86 |
| Recast | **0.99** | 0.86 | 0.75 | 0.87 |
| TildeCNN | **0.99** | 0.92 | 0.81 | 0.91 |
| FastText | 0.97 | 0.91 | 0.76 | 0.88 |
| SemHash | 0.96 | 0.92 | **0.87** | **0.92** |
| BPE | 0.95 | **0.93** | 0.85 | 0.91 |
| HD vectors | 0.97 | 0.92 | 0.82 | 0.90 |

Table 9: Performance of all classifiers for the 20NewsGroups dataset.

| Classifier | $F_1$ **score** | | **Resources: SH vs. HD** | | |
|---|---|---|---|---|---|
| | SH | HD | Train speed-up | Test speed-up | Memory reduction |
| MLP | 0.72 | 0.64 | 53.23 | 79.50 | 93.19 |
| Passive Aggr. | 0.74 | 0.69 | 103.64 | 202.95 | 93.42 |
| SGD Classifier | 0.70 | 0.66 | 105.43 | 186.31 | 93.42 |
| Ridge Classifier | 0.16 | 0.71 | 45.46 | 338.01 | 93.42 |
| KNN Classifier | 0.31 | 0.31 | 184.47 | 65.87 | 127.54 |
| Nearest Centroid | 0.08 | 0.15 | 212.75 | 254.74 | 93.42 |
| Linear SVC | 0.75 | 0.69 | 5.11 | 176.62 | 93.42 |
| Random Forest | 0.58 | 0.26 | 4.27 | 21.43 | 93.41 |
| Bernoulli NB | 0.60 | 0.15 | 57.72 | 56.54 | 93.42 |

