# OpenReview forum: "HyperEmbed:  Tradeoffs Between Resources and Performance in NLP Tasks with Hyperdimensional Computing enabled embedding of n-gram statistics "
_ICLR.cc/2020/Conference — Reject_

### Official Review · AnonReviewer3 · 2019-10-23
**Official Blind Review #3**

**Rating:** 1

**Review:**

This paper shows a trade-off relationship between computational cost (memory usage, train/test time) and performance on several NLP machine learning algorithms that use n-gram statistics. The authors claim that a simple n-gram representation vector with a conventional classifier (MLP, SVC, Naive Bayes...) is computationally efficient.

The large set of experiments on various conventional NLP models and n-gram statistics provide detail information about the trade-off relation between performance and computational cost. My concern is that the computational efficiency of the conventional NLP model is well known to NLP researchers. It would be nice if the authors provide a more persuasive explanation for the importance of this research question.

**Experience Assessment:**

I have published one or two papers in this area.

**Review Assessment: Checking Correctness Of Derivations And Theory:**

I carefully checked the derivations and theory.

**Review Assessment: Checking Correctness Of Experiments:**

I carefully checked the experiments.

**Review Assessment: Thoroughness In Paper Reading:**

I read the paper at least twice and used my best judgement in assessing the paper.

---

> ### Author Response · Authors · 2019-11-06
> **Authors comments to Official Blind Review #3**
>
> We thank the respected reviewer for the efforts in commenting on the paper. We want to comment on some of the statements in the review, as in our opinion, they do not reflect the essence of our contribution.
>
> 1. “The authors claim that a simple n-gram representation vector with a conventional classifier (MLP, SVC, Naive Bayes...) is computationally efficient.”
> Unfortunately, we want to strongly disagree with such an assessment of our claims. We do claim that n-gram statistics is a well-known and useful technique for NLP models. However, the main focus is on the trade-offs between the computational complexity and the performance of the proposed approach compared to the n-gram statistics-based NLP models. The severity of the problem is clearly formulated: The vector representing the n-gram statistics grows exponentially with n.  Our major contribution is the study of the computational/performance trade-offs of n-gram statistics-based NLP models when the n-gram statistics is being embedded using the principles of hyperdimensional computing. The major technical outcome is the performance trade-off study in the case when the size of the embedding vector representing n-gram statistics is fixed. This cannot be referred to as the “conventional NLP model”. The paper confirms the claims by an extensive experimental studies on the set of well-known datasets. The major outcome for the considered tasks is that it was possible to achieve comparable classification performance while reducing memory and time costs by several times compared to the non-embedded (we use the term “conventional” in the paper) n-gram statistics.
>
> 2. “My concern is that the computational efficiency of the conventional NLP model is well known to NLP researchers.”
> We agree that a model based on the non-embedded n-gram statistics is “the conventional NLP model.” However, the experiments in the paper are mostly studying the model with the embedded (mapped) n-gram statistics. The usage of hyperdimensional computing for embedding n-gram statistics in the NLP domain cannot be treated as being a part of “the conventional NLP model” as, to the best of our knowledge, the number of studies in this area is very limited to maximum three other papers.This is the first time where the computational efficiency of the embedded n-gram statistics is studied at all and in such an extensive manner on numerous datasets.
>
> 3. “It would be nice if the authors provide a more persuasive explanation for the importance of this research question.”
> When writing, we assumed that this research question (i.e., the exponential grows of the n-gram statistics vector) is well-known to the NLP community. Compared to the common knowledge about the computational efficiency of “the conventional NLP model”, this paper, to the best of our knowledge, is the first attempt to offer a practical technique (i.e., hyperdimensional computing-based embedding) and its systematic evaluation (e.g., showing possible trade-offs for different classifiers) for improving the computational efficiency. In the revised version, we will add a stronger motivation for highlighting the importance of this research question.
>
> Concluding our commentary, we hope that the provided clarifications will help the respected reviewer to change his/her mind about the depth of the technical contribution of this paper and the level of importance to the NLP community.

---

### Official Review · AnonReviewer1 · 2019-10-24
**Official Blind Review #1**

**Rating:** 3

**Review:**

This paper proposes the use of hyperdimensional (HD) vectors to represent n-gram statistics. The HD vectors are first generated from the whole corpus. Then, it is aggregated or bundled to a vector for each sample as an input of a classifier training. The evaluation is conducted on four datasets: Chatbot, AskUbuntu, WebApplication and 20 News Group using a bunch of classifier including KNN, Random Forest, MLP etc.

It is interesting to see how to hash/project the high dimensional n-gram vector into a lower space for efficiency. The approach is useful in online production systems, and it is eco-friendly. However, there are a few concerns detailed as follows:

1. Can it be generalized to contemporary learned embeddings, e.g., word2vec and GloVe?

2. Lack of proper baselines for comparison. Word2vec, GloVe are trained on large corpora once and can be applied directly to other tasks, and they should be served as baselines. Furthermore, the simple bag of word/TF-IDF should be included as baselines as well.

3. Lack of analysis: it is hard to understand what kind of HD vectors are generated. Are these n-grams semantically related projected nearby in the HD space? This helps readers to understand the constructed embeddings.

4. The SentEval benchmark is popular in sentence level representation learning and it is well known. It is better to see some evaluations on it as well. http://www.lrec-conf.org/proceedings/lrec2018/pdf/757.pdf

Minor comments:
1. The Subword Semantic Hashing is originally from DSSM published in 2013.  (https://www.microsoft.com/en-us/research/wp-content/uploads/2016/02/cikm2013_DSSM_fullversion.pdf)
2. What is $v_c$ in 4.2?


**Experience Assessment:**

I have read many papers in this area.

**Review Assessment: Checking Correctness Of Derivations And Theory:**

I did not assess the derivations or theory.

**Review Assessment: Checking Correctness Of Experiments:**

I assessed the sensibility of the experiments.

**Review Assessment: Thoroughness In Paper Reading:**

I read the paper at least twice and used my best judgement in assessing the paper.

---

> ### Author Response · Authors · 2019-11-07
> **Authors comments to Official Blind Review #1. Part 2**
>
> 3. “Lack of analysis: it is hard to understand what kind of HD vectors are generated. Are these n-grams semantically related projected nearby in the HD space? This helps readers to understand the constructed embeddings.”
> We agree with the reviewer that the analytical part has not been prioritized in the paper. This exclusion is, however, done on purpose as we would like to empirically demonstrate the usefulness of embedding n-gram statistics to HD vectors. Indeed, theoretically, the whole approach works because the embedding is done in such a way that in the projected HD space, two similar n-gram statistics (in the original space) remain still similar. There exist analytical results demonstrating why this is possible. In particular, we know that the bundling operation allows us storing information in HD vectors. There is a recent rigorous work studying this property in the details, but as of now, due to the lack of space, there is only a brief footnote 4, which points the reader to this work. To be more specific, we could extend this footnote in the revised version.
>
> 4. “The SentEval benchmark is popular in sentence level representation learning and it is well known. It is better to see some evaluations on it as well.”
> Thank you for pointing us to the SentEval benchmark. As mentioned above, in the present study, we do not aim at learning superior embeddings for sentences. We instead suggest that whenever n-gram statistics are sufficient for solving a problem, we could do much better in terms of the computational efficiency (plus trade-off) by embedding the n-gram statistics to an HD vector. We think that the currently reported results are sufficient to demonstrate the promise of this claim. However, if the reviewer thinks that reporting the results on this benchmark is compulsory for a successful rebuttal, we are committed to trying our best to perform new experiments with this benchmark. Please let us know.
>
> 5. “Minor comments”
> We thank the reviewer for the careful reading and spotting the minor issues. They are easily fixed in the revised submission.
>
> We hope that this communication clarifies the complications raised by the respected reviewer, such as our design choices, and motivate the choice of the baseline as well as the absence of word embeddings among these baselines.

---

> ### Author Response · Authors · 2019-11-07
> **Authors comments to Official Blind Review #1. Part 1**
>
> We thank the respected reviewer for the encouraging comments on the paper.
> Below we provide our initial comments on the feedback. As of now, these comments have not involved any additional experiments.
>
> 1. “Can it be generalized to contemporary learned embeddings, e.g., word2vec and GloVe?”
> In general, the answer is yes. There is a word embedding technique called Random Indexing. This technique is based on the principles of hyperdimensional computing and, it is data-driven, i.e., the resultant embeddings are learned (more details can be found in, e.g., [1]). One of the advantages of Random Indexing is that it does not require iterative learning as it learns incrementally and, thus, it could be helpful in online production systems. However, since in this paper, we were focusing on the conventional n-gram statistics, which does not require learning, we deliberately were not going into great details of different words embedding techniques. It is also worth recalling that the considered embedding of the conventional n-gram statistics to an HD vector does not require learning as such since it is based on randomly generated HD vectors.
>
> [1] G. Recchia, M. Sahlgren, P. Kanerva, and M. N. Jones, “Encoding Sequential Information in Semantic Space Models. Comparing Holographic Reduced Representation and Random Permutation,” Computational Intelligence and Neuroscience, pp. 1–18, 2015.
>
> 2.1 “Lack of proper baselines for comparison.”
> Our primary claim is that with HD vectors, we can approximate (even accurately) the results obtained with the conventional n-gram statistics. The natural consequence of this claim is that the most proper baseline for classification performance comparison is the conventional n-gram statistics itself. Also, we do not make any definite statements such as that the n-gram statistics is a superior technique for solving NLP problems. We only claim that it is a well-known technique and that it is still a very useful technique for numerous problems.
>
> 2.2 “Word2vec, GloVe are trained on large corpora once and can be applied directly to other tasks, and they should be served as baselines.”
> While designing the evaluation experiments, we were actively discussing whether we should use word embeddings such as Word2vec and GloVe as baselines. Internally, we agreed that from a computational point of view, it would be unfair neglecting the computational resources spent while training these embeddings.  Once we came to this point, considering trainable word embeddings was out of the question as to the resources needed to train them are significantly higher. We made one exception for the case of FastText, which are the trainable subword embeddings. In the paper, however, we have clearly mentioned that we could not compare the training and test time of the technique. This was done in order to remain fair in the evaluation part of the paper and to demonstrate our work with all fairness. We do not argue that this is the only possible judgment on this matter. We want to share with you our motivation when leaving word embeddings aside in the experiments. On top of this, we need quite some memory even to keep the learned embedding for each word in the dictionary. Last, as mentioned above, we do not argue that the n-gram statistics are a silver bullet for NLP problems. In other words, the aim of the paper was not to get the state-of-the-art results but rather improve the time and space complexity associated with the n-gram statistics. Nevertheless, we are ready to compare the classification performance of word2vec and GloVe with our approach if the reasoning above is not convincing.
>
> 2.3 “Furthermore, the simple bag of word/TF-IDF should be included as baselines as well.”
> Regarding the simple bag of word/TF-IDF, our initial assessment was that since the dimensionality of the input feature vector equals the number of words in the dictionary, the computational efficiency of both approaches would not be much better than that of the conventional n-gram statistics. This assumption is correct, at least for Chatbot, AskUbuntu, and WebApplication, where the number of unique n-grams is in the order of several thousand. The additional motivation for not using the raw bag of word/TF-IDF in the experiments was that in the works related to FastText, BPE, and SemHash methods it was argued that subword representations help in getting better performance compared to word-based representations at least for a smaller dataset due to the limited amount of data. Due to this reasoning, we were not considering the bag of word/TF-IDF as proper baselines in the reported results. We, however, are ready to perform the experiments with the raw bag of word/TF-IDF in the revised version of the paper if you think that our reasoning is not fair.

---

### Official Review · AnonReviewer4 · 2019-11-25
**Official Blind Review #4**

**Rating:** 3

**Review:**

This paper introduces a technique to project n gram statistic vectors into a lower dimensional space in order to improve memory efficiency and lower training time. The paper is motivated by the important problem of trying to improve efficiency of existing language models which can be extremely resource intensive. The authors then compare the performance of n gram statistics with HD vectors on 4 datasets to demonstrate that embedding into HD vectors can preserve performance while reducing resource utilization.

	1. The main methodological contribution (using HD vectors) are a nice contribution. I would like the authors to clarify why they chose those 3 operations to operate on vectors for this particular task.
	2. Lack of a competitive baseline : I would like to see how this method compares with existing techniques to speedup n grams such as Pibiri et al. (https://arxiv.org/pdf/1806.09447.pdf). I believe there are other works which work on speeding up n gram models as well but not comparison is presented.
	3. Minor comment: the authors mention using a SGD classifier but I fail to understand what they mean by this. SGD is an optimization technique and not a classifier so I would like the authors to correct this in the paper.

As it currently stands, the lack of strong baselines and the incremental nature of the contribution lead me to believe that this paper does not represent a sufficient advance to warrant publication. I would advise the authors to consider submitting to a more specialized venue (in NLP).


**Experience Assessment:**

I do not know much about this area.

**Review Assessment: Checking Correctness Of Derivations And Theory:**

I assessed the sensibility of the derivations and theory.

**Review Assessment: Checking Correctness Of Experiments:**

I assessed the sensibility of the experiments.

**Review Assessment: Thoroughness In Paper Reading:**

I read the paper at least twice and used my best judgement in assessing the paper.

---

### Decision · Program_Chairs · 2019-12-19

**Decision:**

Reject

**Comment:**

The reviewers were unanimous that this submission is not ready for publication at ICLR in its present form.

Concerns raised included lack of relevant baselines, and lack of sufficient justification of the novelty and impact of the approach.